# Physico-Chemical Characterization and Antimicrobial Properties of Hybrid Film Based on Saponite and Phloxine B

**DOI:** 10.3390/molecules26020325

**Published:** 2021-01-10

**Authors:** Nitin Chandra teja Dadi, Matúš Dohál, Veronika Medvecká, Juraj Bujdák, Kamila Koči, Anna Zahoranová, Helena Bujdáková

**Affiliations:** 1Department of Microbiology and Virology, Faculty of Natural Sciences, Comenius University in Bratislava, 842 15 Bratislava, Slovakia; d1@uniba.sk (N.C.t.D.); matiassko@icloud.com (M.D.); kamila.koci@uniba.sk (K.K.); 2Department of Experimental Physics, Faculty of Mathematics, Physics and Informatics, Comenius University in Bratislava, 842 48 Bratislava, Slovakia; veronika.medvecka@fmph.uniba.sk (V.M.); anna.zahoranova@fmph.uniba.sk (A.Z.); 3Department of Physical and Theoretical Chemistry, Faculty of Natural Sciences, Comenius University in Bratislava, 842 15 Bratislava, Slovakia; uachjuro@savba.sk; 4Institute of Inorganic Chemistry of SAS, 845 36 Bratislava, Slovakia

**Keywords:** *Staphylococcus aureus*, clay mineral, photosensitizer, hybrid film, antimicrobial

## Abstract

This research was aimed at the preparation of a hybrid film based on a layered silicate saponite (Sap) with the immobilized photosensitizer phloxine B (PhB). Sap was selected because of its high cation exchange capacity, ability to exfoliate into nanolayers, and to modify different surfaces. The X-ray diffraction of the films confirmed the intercalation of both the surfactant and PhB molecules in the Sap film. The photosensitizer retained its photoactivity in the hybrid films, as shown by fluorescence spectra measurements. The water contact angles and the measurement of surface free energy demonstrated the hydrophilic nature of the hybrid films. Antimicrobial effectiveness, assessed by the photodynamic inactivation on hybrid films, was tested against a standard strain and against methicillin-resistant bacteria of *Staphylococcus aureus* (MRSA). One group of samples was irradiated (green LED light; 2.5 h) and compared to nonirradiated ones. *S. aureus* strains manifested a reduction in growth from 1-log_10_ to over 3-log_10_ compared to the control samples with Sap only, and defects in *S. aureus* cells were proven by scanning electron microscopy. The results proved the optimal photo-physical properties and anti-MRSA potential of this newly designed hybrid system that reflects recent progress in the modification of surfaces for various medical applications.

## 1. Introduction

Healthcare-associated infections (HAIs) are observed in several countries in Europe [1,2], the USA [3], UK [4], and Australia [5]. These HAIs have caused financial harm, as well as morbidity and mortality of patients all over the world [6,7]. HAIs are strongly associated with biofilm-forming microorganisms on medical devices such as central venous catheters [8], urinary catheters [9], orthopedic implants [10], and dental implants [11]. The Gram-positive bacterium *Staphylococcus aureus* is currently assumed to be the most frequent pathogen associated with HAIs [12,13,14].

Numerous approaches have been designed to prevent colonization or to eradicate microorganisms from the surface. Disinfection with quaternary alkylammonium salts, hydrogen peroxide, or sodium hypochlorite has been in use for many years [15], while the surface modification of medical equipment with nanoparticles is currently in high demand [16,17,18]. Additionally, nanoparticles are good carriers for antimicrobial drugs; for example, gentamicin carried by silica nanoparticles [19] or poly(ethylene)glycol-poly(β-amino esters) micelles conjugated with triclosan demonstrated an effective reduction in Gram-positive as well as Gram-negative bacteria [20]. Some nanoparticles such as the carboxyl-grafted superparamagnetic iron oxide nanoparticles were applied directly for disruption of biofilms [21]. Photodynamic inactivation (PDI) is another strategy showing great potential in biofilm prevention or killing [22,23]. However, the above-mentioned strategies, when used separately, have limitations in their effectivity, longevity, and bioavailability [24,25]. Using a combination of the above approaches is the least explored area, but has immense potential. The idea of this study was to design a hybrid system based on nanoparticles of clay minerals with an immobilized photosensitizer (PS) that could be used for PDI [26]. PDI is an approach employing nontoxic PS, irradiated with light of an appropriate wavelength, resulting in the formation of reactive oxygen species (ROS) that destroy the structure of microbial cells [27,28]. Previously, some PSs proved to be effective in PDI against *S. aureus*, for example, natural green colorant E-141, toluidine blue bound to chitosan, microemulsion, and also mesoporous silica nanoparticles [29,30,31] or malachite green incorporated into carbon nanotubes [32]. Methylene blue associated with silica nanoparticles as a carrier has also demonstrated antimicrobial activity [33].

In hybrid systems, the nanocarrier is an important component that not only provides delivery but also the conditions for the PS to act effectively. Our nanomaterial was based on the synthetic layered silicate saponite (Sap), which has already been described as an optimal vehicle for different molecules [34,35]. Moreover, Sap has also manifested antimicrobial properties against Gram-negative bacteria of *Escherichia coli* [36]. The newly designed hybrid system contained phloxine B (PhB) in the role of PS. This compound is a commercially available dye used in cosmetics, veterinary drugs, medicinal drugs, and a food colorant, generally known as D&C Red no. 28, approved by the Food and Drug Administration [37,38]. PhB manifests suitable photoactive properties, absorbing visible light at 540 nm. Moreover, PhB was effective in PDI in the reduction of oral plaques caused by *Streptococcus mutans* [39] and *Bacillus subtilis* [40]. As it has anionic functional groups, it does not adsorb on layered silicates directly. However, stable hybrid systems can be synthesized, starting from modified materials [41]. The objectives of this work included the preparation and characterization of so-called organoclays prepared by modifying Sap with quaternary alkylammonium cations and functionalizing it with PhB molecules. The main goal was to characterize the antimicrobial properties of these materials using PDI. Modification with quaternary ammonium cations should lead to the efficient adsorption of anionic dyes [42]. On the other hand, organic anions should bind less strongly [43], which may affect the interaction of PhB anions with the cells of the microorganism and the antimicrobial effect of the material. This study: i) introduces a hybrid system based on a layered silicate and a PS useful for PDI; ii) provides its photo-physical characterization; iii) describes its antimicrobial activity against resistant bacteria of *S. aureus.* This research presents a newly designed hybrid system with antimicrobial properties, potentially useful for different medical applications.

## 2. Results and Discussion

The growing adverse impact of biofilm-related infections accompanied by resistance against a wide spectrum of antibiotics has made clear the importance of developing novel materials with antimicrobial properties [44,45], which was the main motivation for the preparation of a hybrid material based on the silicate with immobilized PhB. In preliminary experiments, several silicate materials were tested to select the silicate sample with optimal properties. Colloidal stability, sufficient adsorption of PS, and transparency of the hybrid films were the main criteria for the selection. The layered silicate Sap was chosen, because it is easy to delaminate and exfoliate into nanolayers in water. After the modification and functionalization with PS, it formed optically homogeneous and transparent films (see discussion below). This silicate, with its parameters such as particle size and layer charge, meets the criteria for the modification of polymer nanocomposites and controlled drug delivery [46,47,48]. Its layer charge can provide a static anchor to help with the proper distribution and assembly of different molecules, including those with photoactive properties [44,49], which was also confirmed in this study. As shown below, prepared hybrid films manifest suitable physico-chemical properties, but also antimicrobial effectiveness using PDI.

### 2.1. Physico-Chemical Characterization

The prepared hybrid material underwent detailed characterization in terms of structure, and especially photo-physical properties. XRD patterns of the films were measured for structural characterization, particularly to determine the distance between the layers due to the intercalation of the surfactant and PhB molecules (Figure 1). The film of Sap representing the starting material for functionalized films exhibited a relatively broad diffraction peak, which can be attributed to the small particle size of this smectite (diameter ~ 20 nm). The *d*_001_ value of the Sap film is essentially in agreement with the values of similar smectite samples. From this value, it is possible to estimate the average distance between the opposing layers as being approximately 0.3 nm, which represents one layer of hydrated Na^+^ cations and water molecules filling the space between them.

Organoclays were prepared by saturating the Sap film with the excess of surfactant solutions of various concentrations. Two types of such films were prepared, designated O_2.0_S and O_1.0_S. These films were prepared by saturating the film of Sap with surfactant solutions of concentrations of 2 and 1 mmol/L, respectively. Both the films were modified with an excess of the surfactant over the cation exchange capacity of Sap (0.72 mmol g^−1^). The excess of ODTMA bromide over cation exchange capacity (CEC) was removed by washing with isopropanol. The films of both types should have very similar properties and essentially the same amounts of the intercalated surfactant cations. The modification of Sap with organic cations resulted in a significant increase in the parameter *d*_001_ to 1.83–2.07 nm. These values indicate the presence of trimolecular layers formed by the long alkyl chains of ODTMA. With the O_1.0_S film, the intercalated phase may also contain a certain proportion of bimolecular layers. This assumption would also be consistent with a partially larger peak width for the O_1.0_S film, although the differences were minimal. Possibly a small part of the surface of O_1.0_S was not completely modified. Both film types represented organoclays with a good ability to adsorb the anionic dye, PhB. The dye adsorption led to a change in the color of the film, which could be observed visually. It should be emphasized here that Sap alone is not able to adsorb this dye. The XRD patterns did not change significantly with the dye intercalation (films PO_2.0_S and PO_1.0_S), due to the relatively small amounts of the adsorbed dye compared to the amount of intercalated ODTMA cations.

Tens of such films had to be prepared for microbiology experiments. There needed to be a reproducibility of their properties and quality in terms of the homogeneous distribution of the components in the films. In part, a poorer reproducibility of the properties of the PO_1.0_S film was observed, as well as higher heterogeneity in the dye distribution detected by optical microscopy (not shown). This was probably due to the insufficient modification of the Sap film due to the lower concentration of surfactant solution used in the preparation of O_1.0_S-type films. Although the added amount of the surfactant seems to be sufficient with respect to the cation exchange capacity of Sap, the surfactant molecules may not have penetrated sufficiently inside the films. A slight difference between basal spacing values of the O_1.0_S and O_2.0_S is already indicated in the XRD patterns (Figure 1). An insufficient modification may have led to a nonhomogeneous functionalization of the surface with PhB molecules during the preparation of the PO_1.0_S film.

The basic method for characterizing optical properties was absorption spectroscopy in the visible region (Figure 2). The spectra of the films of both types functionalized with PhB consist of the main band at 556 nm, which corresponds to the π–π* transition, and a vibronic component at 525 nm of the dye. The absorption spectra of the films indicate that the spectral properties of PhB were only partially changed compared to the solution, and had similar properties to the films of hybrid materials based on Sap modified with a cationic polymer [41]. The Sap film and the corresponding organoclay were also measured as blank samples without PhB, and did not absorb light in the visible region. The recorded spectra of blanks showed baselines with a slight increase in absorbance with light energy due to light scattering. Functionalized films containing PhB exhibited even higher baselines, which indicate the presence of structural defects that caused a higher light scattering. Such changes probably occurred during the immersion of the films in the PhB solution. Nevertheless, it is possible to estimate the dye concentration in the films by using the maximum absorbance and subtracting the values of the baseline at this wavelength. As films are characterized by surface concentration (*Γ*), the calculation should be derived from a general form of the Lambert–Beer law:(1)$$A=\varepsilon \int_{0}^{l}c\left(z\right)dz$$

If the film thickness is neglected, the simplification considers the presence of all dye molecules on the planar surface:(2)$$A=\varepsilon^{\prime }\Gamma_{\mathrm{PhB}}=10^{3}\varepsilon_{\mathrm{PhB}}\Gamma_{\mathrm{PhB}}$$
where $\varepsilon^{\prime }$ represents the molar absorption coefficient expressed in mol^−1^ cm^3^ cm^−1^ units to be compatible with the surface concentration expressed in mol cm^−2^. The molar absorption coefficient *ε*_PhB_ = 8.3 × 10^4^ mol^−1^ dm^3^ cm^−1^ is taken from the literature [50]. *Γ*_PhB_ can be calculated according to the equation:(3)$$\Gamma_{\mathrm{PhB}}=\frac{A}{10^{3}\varepsilon_{\mathrm{PhB}}}=\frac{10^{\;-\;3}A}{\varepsilon_{\mathrm{PhB}}}$$

The films had an area of 4.84 cm^2^ containing 0.6 mg Sap, which represents 0.124 mg cm^−2^. The surface concentration of the dye in the films (*Γ*_PhB_) can be estimated using Equation (3), the measured absorbance from the maxima of the PhB bands, and considering the value of *ε*_PhB_. The values *Γ*_PhB_ were estimated to be 1.711 × 10^−6^ and 1.871 × 10^−6^ mmol cm^−2^ for PO_2.0_S and PO_1.0_S, respectively. The amount of dye adsorbed per unit mass of Sap was 1.38 × 10^−2^ and 1.51 × 10^−2^ mmol g^−1^ in PO_2.0_S and PO_1.0_S, respectively, which in both the films, is below 2% of the CEC value. The difference between these two films was negligible.

The emission spectra of representative films with adsorbed PhB (PO_2.0_S; PO_1.0_S) were measured to verify that the intercalated PhB molecules retain their photoactivity. The films were found to exhibit relatively strong fluorescence (Figure 3A). The emission wavelength shifted to partially higher wavelengths compared to the solution [50] and a similar Sap-based material modified with a cationic polymer [41]. Emission spectra were measured using excitation at different wavelengths (not shown) to verify whether the samples contained one or more fluorescent species. Only selected representative spectra measured using two excitation wavelengths are shown (Figure 3A). It is clear that the change in excitation wavelength only led to changes in the intensity of fluorescence, and no change in the shape of the emission bands and no spectral shifts occurred. This proves that, despite the heterogeneous nature of the material, there is only a single spectral form present with very similar characteristics for both films. The presence of a single form of PhB and the observed strong fluorescence reveal some important properties of the materials. The organoclay is not only suitable for the adsorption of the anionic PhB, but can also completely suppress dye molecular aggregation. The presence of molecular aggregates would result in spectral complexity, fluorescence quenching, and a reduction in dye photoactivity. Hybrids with photoactive dye molecules are also a basic prerequisite for the activity of the material in terms of photosensitization. The assumptions made based on the emission spectra were also confirmed by the excitation spectra (Figure 3B). The excitation spectra match well with the profiles of corresponding absorption spectra (Figure 3), suggesting that the structure of the PhB molecules does not change significantly in the excited state, which may be an important indicator for the stability of the dye molecules in the films. The PO_2.0_S film gave a slightly higher emission in the excitation spectra than PO_1.0_S (Figure 3B), which was similar to the trend observed in the emission spectra (Figure 3A). A relatively larger interlayer expansion was observed in the O_2.0_S film (see *d*_001_ values, Figure 2). The homogeneous and complete modification of Sap by ODTMA cations and higher interlayer expansion in the O_2.0_S film may contribute to better photo-physical properties of absorbed PhB. The O_1.0_S film, which was prepared by intercalating the surfactant at a lower concentration, adsorbed the dye relatively inhomogeneously on its surface, which was observed by optical microscopy of the PO_1.0_S film (not shown). On the other hand, the excitation spectra recorded from different sites on the PO_1.0_S film (near the center and edge of the slide) exhibited almost identical intensities of emitted light (Figure 3B, dashed lines). This means that the differences in the spectra of the films being compared are not due to macroscopic film inhomogeneities and variable distribution of the adsorbed dye on the surface. This could inadvertently lead to different emission intensities and heterogeneity on the microscopic scale. However, the presence of microscopic inhomogeneity is not suitable for microbiological testing. Therefore, due to some differences related to the higher reproducibility of the parameters, and higher luminescence of the PO_2.0_S film, biological experiments continued only with the PO_2.0_S films.

The hydrophobicity of a surface is generally expressed in terms of contact angle. This method evaluates the wettability of the surface and enables the dispersive and polar SFE components to be determined. Both films were tested for hydrophobic/hydrophilic properties, as they are critical for microbial adhesion and can promote biofilm development. For example, a nanocomposite film based on *Salvia macrosiphon* seed mucilage and Cloisite 15A (montmorillonite) modified with a quaternary alkylammonium salt exhibited significant activity against *S. aureus* associated with increased hydrophobicity in the presence of the nanoclay [51]. On the other hand, the hydrophilicity of the material provides lubricity and slippery surfaces under wet conditions, which are crucial in the design of, for example, central venous catheters. This also offers conditions for embedding active reagents in these surfaces that help in their controlled release [52,53]. However, not only the surface of the material, but also the cell surface hydrophobicity of bacterial strains is critical, as bacteria with higher cell surface hydrophobicity colonized the hydrophobic surface at a higher rate, and bacteria with a hydrophilic cell surface form a better biofilm on more hydrophilic surfaces [54,55]. The results summarized in Figure 4 show the highly hydrophilic nature of: (A) Sap, and the lower hydrophilic properties of the (B) PO_1.0_S and (C) PO_2.0_S films of organoclays functionalized with PhB. These films exhibited a higher SFE compared to the control sample based on the Sap film. However, the PO_1.0_S film exhibited a nonhomogeneous surface leading to a higher error rate in their SFE. The irregular distribution of ODTMA on the material caused an inconsistency in the repeatability of the material’s preparation. On the other hand, the PO_2.0_S film exhibited a low error rate in SFE, as it possesses an even, homogenous distribution of the ODTMA over the whole cover glass, resulting in a consistent hybrid film that was finally used for testing anti-biofilm effectiveness.

### 2.2. Antimicrobial Activity of PDI

Prior to testing antimicrobial properties, MRSA strains were confirmed by PCR determining the presence of the *Mec A* gene (Appendix A). A preliminary experiment with PhB suggested an optimal concentration of this agent that was later used for the preparation of hybrid films. The optimal concentration took into account the maximal difference in the reduction of bacterial growth after PDI compared to the conditions without irradiation. Moreover, a sample without irradiation had to manifest a minimal reduction in growth compared to the sample with the Sap film alone, as is documented in the paragraph below. The results are summarized in Figure 5. The concentrations of 0.001, 0.005, and 0.01 mM PhB did not exhibit a significant difference in growth rate compared to the control group (growth rate over 10^10^ CFU/mL) under both dark and light for all three strains. The effectiveness of 0.05 and 0.1 mM PhB resulted in about 4 log_10_ reductions in growth under irradiation with all three isolates, proving a very strong antimicrobial effectiveness (*p* < 0.05). However, 0.1 mM PhB resulted in a significant reduction even in the dark, while 0.05 mM PhB did not produce a reduction in the dark, so this concentration was finally selected for the preparation of the hybrid film. These results are in agreement with research conducted by Rasooly and Weisz, who assessed the effect of PhB on MRSA bacteria. They proved that 25 µg/mL (0.03 mM) is the minimal inhibitory concentration, as this concentration inhibited 90% of *S. aureus* growth, and at concentrations of 50 (0.06 mM) or 100 µg/mL (0.12 mM), bacterial growth stopped completely with a reduction in CFU by 10^4^ within 3 h with standard room illumination [56]. It was also possible to obtain this antimicrobial effectiveness with our samples (data not shown), as the killing effect proved to be concentration-dependent, but because of a study of the PDI process, a lower concentration of PhB was selected for film preparation.

### 2.3. Anti-Biofilm Effectiveness of PDI with Hybrid Film

The hybrid films were prepared according to the protocol described in Scheme 1. Antimicrobial activity was estimated against all three *S. aureus* strains. Pilot experiments indicated better reproducibility of the results of biological experiments with PO_2.0_S-type films, which might be related to some properties of the tested films as discussed above. The bacterial growth rate was very high (10^7^–10^9^ CFU/mL) under both dark and light on a Sap-coated cover glass (control group), while the PO_2.0_S film only exhibited slightly decreased growth compared to the Sap control group in the dark. However, the strains of *S. aureus* CCM 3953, L12, and L18 exhibited reductions of 3log_10_, over 3log_10_, and over 2log_10_, respectively, after irradiation (*p* < 0.05; Figure 6). These results demonstrated antimicrobial activity of the PO_2.0_S film but showed diversity in the growth rate of tested strains, suggesting differences in biofilm-forming capacity. On the other hand, efficiencies on MRSA strains suggested that the effect of the PO_2.0_S film is not associated with MRSA phenotype, but is strain-specific [57].

### 2.4. SEM of S. aureus Biofilm on the Hybrid Films

The micro-photographs summarized in (Figure 7) only show a thick adherence of *S. aureus* CCM 3953 on the film with Sap (A); the PO_2.0_S film in the dark also exhibited a high adhesion of bacteria on the surface (B); whereas after irradiation, the growth of bacteria was radically decreased (C). In clinical strain L 12, the adherence to the Sap (D) was lower than to the standard strain. In the dark, the growth of bacteria (E) was similar to that on the Sap alone. However, after irradiation, the growth of the L12 strain was significantly reduced, indicating anti-biofilm activity of PDI (F). Moreover, the details in Scheme 1 show disruption of the cell wall of bacteria compared to samples G and F, suggesting that bacteria were unable to recover. These results are in agreement with the previous effectiveness of the PO_2.0_S film in PDI. The figure also proved the different adherence capacity of both strains that could be associated with different abilities to form a biofilm.

## 3. Material and Methods

### 3.1. Bacterial Strains and Antimicrobial Effectiveness of PhB

The standard strain *S. aureus* CCM3953 (Czech Collection of Microorganisms, Brno, Czech Republic) and 2 methicillin-resistant *S. aureus* (MRSA) L12 and L18 were tested in this study. Both clinical isolates were acquired from central venous catheters (Institute of Microbiology, Faculty of Medicine, Comenius University in Bratislava, and University Hospital Bratislava, kindly provided by prof. Lívia Slobodníková, Ph.D.). Strains were cultured on Mueller Hinton Agar (MHA, Biolife, Milan, Italy) at 37 °C for 24 h. The conditions for PCR and the set of primers specific for the *MecA* gene (forward: GTA GAA ATG ACT GAA CGT CCG ATA A; reverse: CCA ATT CCA CAT TGT TTC GGT CTA; Metabion International AG, Germany) were previously described by Braoios and Flumhan [58](details are described in Appendix A).

The preliminary testing determined the antimicrobial activity of PhB against the standard strain, as well as resistant isolates of *S. aureus*. A 10 mM stock solution of PhB (Sigma-Aldrich, St. Louis, MO, USA) was prepared in deionized water, filtered through a 0.22 µm filter (TPP, Trasadingen, Switzerland) and stored at 4 °C in the dark for, at most, 2 weeks. The experiment was performed in 96-well flat-bottom polystyrene plates (Sarstedt, Germany), which were filled with 100 µL/well of MHB with bacteria (≈2 × 10^7^ cells/mL) and 100 µL of MHB with PhB to obtain final concentrations of PhB as follows: 0.001, 0.005, 0.01, 0.05, and 0.1 mM. The assay was performed both in the dark and with irradiation. The plates were sealed with parafilm and pre-incubated in the dark for 1 h in an incubator at 37 °C without shaking (Thermostatic cabinet, Lovibonds, Biosan, Riga, Latvia). Then, one set of the plates was irradiated with green LED light (2.42 mW·cm^−2^) positioned 6 cm above the plate for 2.5 h. The control samples contained only MHB with strains incubated under the same conditions as the irradiated and nonirradiated samples. The nonirradiated samples were covered with aluminum foil. After irradiation, the incubation continued for another 24 h in the dark at 37 °C. Then, cells were scraped from the wells, 10-fold serially diluted in phosphate-buffered saline (PBS; 137 mM NaCl, 2.7 mM KCl, 8 mM Na_2_HPO_4_, and 2 mM KH_2_PO_4_; CentralChem, Bratislava, Slovakia), then plated on MHA and incubated for 24 h at 37 °C. Antimicrobial activity was determined by counting the number of colony-forming units (CFU), which were then calculated for 1 mL. The results from irradiated and nonirradiated samples were compared to the control sample containing Sap alone. The assay was performed in 3 parallels, and the experiments were repeated twice. Results are expressed as average values with standard deviations.

### 3.2. Preparation of Hybrid Film

The preparation of the hybrid film system is summarized in Scheme 1. Briefly, the cover glasses (thickness 0.4 mm, 22 × 22 mm; Thermo Scientific, Germany) were immersed in piranha solution (3:1 of sulfuric acid and hydrogen peroxide; CentralChem, Bratislava, Slovakia) for 30 min, and thoroughly washed with distilled water before being dried at room temperature (RT). The Sap (Kunimime Ind., Tokyo, Japan) was resuspended in deionized water at a concentration of 1 g/L, stirred, and sonicated with a sonicator at 55 kHz (Branson 200 ultrasonic cleaner, Danbury, CT, USA) for 10 min. The 600 µL of Sap was evenly coated onto cover glasses to allow it to dry overnight at RT. For the surface modification, octadecyltrimethylammonium (ODTMA) bromide (Sigma-Aldrich, Darmstadt, Germany) dissolved in deionized water was stirred and heated until completely dissolved. The Sap-coated cover glasses were transferred into 6-well polystyrene plates (Sarstedt, Nümbrecht, Germany), and 3 mL of one of the surfactant solutions was added. Then, 1 or 2 mM ODTMA bromide solutions were used for the preparation of the films. The added amounts of the surfactant represented values close to the cation exchange capacity of the silicate (1 mmol g^−1^) or its excess (2 mmol g^−1^), respectively. The names of organoclay films O_2.0_S and O_1.0_S express their composition (O-organic cation, S-saponite) and the amount of added organic cations (2 and 1 mmol g^−1^, respectively). The Sap films immersed in the ODTMA solutions (3 mL) were incubated without shaking for 12 h at 40 °C (Thermo shaker PST-60H2-4 Biosan, Riga, Latvia). The cover glasses were then washed with 2-propanol (CentralChem, Bratislava, Slovakia) for 1 min to remove the excess of unbound ODTMA and further washed in deionized water. After washing, the slides with the deposited organoclay films were transferred to a hot-air oven (60 °C for 30 min) for drying. The dried films of organoclays were incubated in 6-well polystyrene plates immersed in 3 mL 0.05 mM PhB solution overnight in the dark at RT. The following day, cover glasses with the films with adsorbed PhB were washed with deionized water to remove the excess unbound dye, dried at RT, and stored in the dark until used. The films functionalized with PhB are named expressing their identity PO_2.0_S and PO_1.0_S, where P is the symbol of phloxine B.

### 3.3. Characterization of Hybrid Film

Representative films were investigated using X-ray diffraction (XRD). The samples were recorded with an EMPYREAN system (PANalytical) for a 2*Θ* range between 2° and 10° to characterize the basal spacing of the sample. A generator was set to 40 mA and 45 kV, using CuK_α_ radiation (0.15406 nm). Absorption spectroscopy in the visible region was investigated using a double-beam Cary 5000 UV-Vis-NIR Spectrometer (Agilent, Santa Clara, CA, USA). Liquid samples and colloidal dispersions were measured using 10 mm quartz cuvettes transparent to UV and visible light (Hellma Analytics, Mullheim, Germany). Steady-state fluorescence measurements were performed using a Fluorolog-3 spectrometer (Horiba Jobin-Yvon, Kyoto, Japan) in front-face setup and using a J1933 Solid Sample Holder from Horiba. The ranges of the measurements, excitation, and emission wavelengths are specified below.

Measurement of the sessile droplet contact angle and determination of the surface free energy were done with a Drop Shape Analyzer—DSA 30 with DSA1 software (Kruss, Hamburg, Germany). The surface free energy (SFE) was quantified by the Owens–Wendt method [59]. For measurement, the two optimal testing liquids were chosen; highly polar distilled water and nonpolar diiodomethane (Sigma-Aldrich, Germany). The volume of droplets was 2 µL. The obtained value was statistically processed from 10 droplets of each liquid measured on the surface of the sample.

### 3.4. Photodynamic Anti-Biofilm Effectiveness

The cover glasses with Sap film (control) and with a film containing a hybrid system were transferred to the 6-well polystyrene plates and sterilized under UV light for 30 min (Esco class 2 BSC, Singapore). Bacterial strains were prepared as already described for testing PhB. Then, 1.5 mL of bacterial suspension in MHB (2 × 10^7^ cells/mL) was seeded onto cover glasses placed in the 6-well polystyrene plates. An additional 1.5 mL of MHB was added to obtain a total volume of 3 mL. The plates were pre-incubated in the dark at 37 °C for 2 h without shaking in the incubator. Then, the glass slides were transferred to new 6-well polystyrene plates containing 3 mL of new MHB. Plates were further processed to the irradiation step, while nonirradiated samples were covered with aluminum foil, but kept under the same conditions as the irradiated ones. Irradiation was performed as was described above. After irradiation, both irradiated and nonirradiated plates were incubated for another 24 h at 37 °C in the dark. The control sample with Sap alone was prepared in the same way, as was the nonirradiated sample with the hybrid system. The following day, all cover glasses were carefully washed with distilled water and dried with sterile cotton swabs to remove adhered bacteria from the bottom side. The cover glasses were then transferred to 50 mL sterile falcon tubes containing 10 mL of 1*PBS and sonicated for 5 min to remove cells bound to the film. These samples were then 10-fold serially diluted, plated onto MHA plates, and incubated at 37 °C. After a 24 h cultivation, the number of CFU was counted. The number of CFU/mL in the irradiated and nonirradiated samples was compared to the control film with Sap alone. Each material was tested in 3 parallel samples, and the experiment was repeated twice. Results are expressed as average values with standard deviations.

### 3.5. Scanning Electron Microscopy of S. aureus Biofilm

The samples for scanning electron microscopy (SEM) were prepared as described above. The bacterial biofilm removed from the film was transferred to new 6-well polystyrene plates and fixed in 4% paraformaldehyde (Sigma-Aldrich, St. Louis, MO, USA) for 1 h in the dark. These samples were then washed twice with 1*PBS and sterile-distilled water for 10 min each at RT before they were post-fixed in 1% osmium tetroxide (OsO_4_) for 1 h in the dark. These samples were again washed twice in 1*PBS and sterile-distilled water for 10 min each, followed by dehydration steps: Washing in 25%, 50%, 75%, and 95% ethanol for 10 min each and twice in 100% ethanol for 15 min. Upon dehydration in 100% ethanol, the samples were allowed to dry at RT. The fixed samples were then sputter-coated with carbon (20 nm) using a QISOT ES Sputter Coater (Quorum Technologies, Lewes, UK) and held with carbon tape on an SEM sample holder and viewed under a JSM-7100F electron microscope (JEOL, Tokyo, Japan).

### 3.6. Statistical Analysis

Values are means from at least 3 parallel and 3 separate experiments. Bars are standard deviations (SDs). The statistical comparison between samples was performed using a one-way analysis of variance (ANOVA). Differences were considered statistically significant at *p* < 0.05.

## 4. Conclusions

The results presented here demonstrated the effective use of synthetic Sap and harmless dye PhB, which is commonly used as a colorant for drugs, cosmetics, and food, for the preparation of hybrid films. They also shed light on the combination of different techniques, in this case, surface modification and PDI for antimicrobial activity against clinically relevant bacteria of *S.aureus*. The results also demonstrate the vital role of the physico-chemical properties of designed hybrid films that generated reliable results in terms of controlling biofilms. In this system, Sap helps with the adherence and modification of surface properties, as well as carrying the PhB molecules. The hybrid film also demonstrated anti-biofilm properties in PDI against the clinically important pathogen of *S. aureus*. This novel hybrid system possesses promising potential and reflects recent progress in the modification of surfaces for various medical applications.

## Data Availability

The data presented in this study are available on request from the corresponding author.

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
