# Peer review of "Physico-Chemical Characterization and Antimicrobial Properties of Hybrid Film Based on Saponite and Phloxine B"

_molecules, 2021, doi:10.3390/molecules26020325_

Round 1

Reviewer 1 Report

In this article the authors discussed on the physico-chemical characterization of hybrid films prepared on saponite and phloxine B. The films are also characterized for their antimicrobial properties.

the manuscript is well written and all the information are reported in details. 

I recommend the publication after some corrections that to me can be useful to improve the paper.

  1. the authors mention several times the "high" performances of the film. In particular at line 52 they say "immense potential", at line 64 "excellent vehicle", again at line 415 "excellent potential". to me, considering the illustrated results, "excellent potential" is a big overstatement. maybe "good" or "interesting" are more appropriate terms.
  2. line 195-207. the first part of the "results and discussion" sounds like an "introduction". no results are discussed here and maybe they can move it in the first section of the manuscript
  3. line 360-366. this part is not clear to me. I think that the authors should better discuss their results with respect to the literature, maybe in a summarizing table with related references.

Author Response

Reviewer general comment:

In this article the authors discussed on the physico-chemical characterization of hybrid films prepared on saponite and phloxine B. The films are also characterized for their antimicrobial properties.

the manuscript is well written and all the information are reported in details. 

I recommend the publication after some corrections that to me can be useful to improve the paper.

Response to the reviewer:

Many thanks to the reviewer. We believe his/her comments contribute to the improvement of the manuscript.

Point 1. Reviewer comment:

the authors mention several times the "high" performances of the film. In particular at line 52 they say "immense potential", at line 64 "excellent vehicle", again at line 415 "excellent potential". to me, considering the illustrated results, "excellent potential" is a big overstatement. maybe "good" or "interesting" are more appropriate terms.

Response to the reviewer:

We followed the advices of the reviewer and changed the expressions as suggested by the reviewer.

Point 2. Reviewer comment:

line 195-207. the first part of the "Results and discussion" sounds like an "Introduction"; no results are discussed here and maybe they can move it in the first section of the manuscript

Response to the reviewer:

Thank you for this comment. We wanted to highlight the importance of our work in connection with already published knowledge. However, maybe this introduction of the results section was too general. Therefore, we have radically changed this part and explained in detail, how an optimal system was selected in the view of preliminary experiments.

Point 3. Reviewer comment:

line 360-366. this part is not clear to me. I think that the authors should better discuss their results with respect to the literature, maybe in a summarizing table with related references.

Response to the reviewer:

This text was modify to clearly distinguish which part of the text relates to our results and which results were obtained by another study that was just compared with ours. We do not think that the summarizing table is necessary, as a number of references covering a participating of phloxine B in PDI is very limited.   

Reviewer 2 Report

This paper describes the synthesis  and characterization of hybrid films based on saponite and phloxine B and the evaluation of their antimicrobial properties using photodynamic therapy. The paper is interesting, and they have performed a thorough characterization of their material; however, there are several issues that need to be addressed before the manuscript is ready for publication. My recommendation is that the manuscript deserves a major revision in its current form and some sections should be expanded.

If the authors are prepared to undertake the work required, I would be pleased to reconsider my decision, although this does not imply that the paper will be accepted with certainty in the event that the authors decide to revise and resubmit the work. 

  1. According to the general editor rules, authors for whom English is not their native language are encouraged to have their paper checked before submission for grammar and clarity. This manuscript needs a revision by a native speaker.
  2. There are many other papers in which a photosensitizer is encapsulated withing polymeric or inorganic matrices. Those are not described in the introduction section. I do not see clearly the advantage of using clay compared to other inorganic matrices (e.g., mesoporous silica, zeolites, etc.) to host a photosensitizer. They mention some examples but I do not clearly see the advantage of using saponite. They claim that one advantage would be its antimicrobial nature (include ref 35). But it is not clearly explained.
  3. The reduction of the photobleaching due to the intercalation of the photosensitizer in the clay is something to look into it. Does the clay reduce the photobleaching?
  4. If the interaction between the modified clay and the photosensitizer is merely electrostatic, only one monolayer of photosensitizer would be adsorbed (bimolecular layers on both surfaces). Does the clay present micropores? Can the photosensitizer be adsorbed on the micropores?
  5. The main limitation I observe is the limited reduction in the cellular viability after long irradiation times (2.5h) also the toxicity of the PhB itself does not justify the need of using light. It is not clear why you need to apply light when the photosensitizer by itself already shows large antimicrobial action. The benefit shown is reduced.
  6. Do the values reported for the contact angles of sapionite match others reported in the literature?
  7. In the photodynamic antibacterial studies as control, they should have used sapionite modified with ODTMA bromide solutions also to decouple the cytotoxic effect of the ODTMA bromide from the cytotoxic effect of the PhB.
  8. A clear dose-dependent effect is not observed for the illuminated samples in Figure 6. They should also test a concentration above 0.1mM.

Author Response

Reviewer general comment:

This paper describes the synthesis  and characterization of hybrid films based on saponite and phloxine B and the evaluation of their antimicrobial properties using photodynamic therapy. The paper is interesting, and they have performed a thorough characterization of their material; however, there are several issues that need to be addressed before the manuscript is ready for publication. My recommendation is that the manuscript deserves a major revision in its current form and some sections should be expanded.

If the authors are prepared to undertake the work required, I would be pleased to reconsider my decision, although this does not imply that the paper will be accepted with certainty in the event that the authors decide to revise and resubmit the work.

Response to the reviewer:

We would like to thank the reviewer for time spending with peer review of the manuscript. We believe, we could respond to his/her comments and modify the manuscript as much as possible to improve its quality.

Point 1 Reviewer comment:

According to the general editor rules, authors for whom English is not their native language are encouraged to have their paper checked before submission for grammar and clarity. This manuscript needs a revision by a native speaker.

Response to the reviewer:

English language corrections have already been done (in original submitted version) by an international company, with which we have been cooperating for many years. The owner of that company is Mr. Ben J. Watson-Jones (native speaker, and moreover, with an education in biochemistry, thus he is able to correct a research text in biochemistry, biology and related fields). We have never had problems with accepting of his English by international journals.

Point 2 Reviewer comment:

There are many other papers in which a photosensitizer is encapsulated withing polymeric or inorganic matrices. Those are not described in the introduction section.

Response to the reviewer:

We added citation mentioned that topic; Introduction p.2, line 58 (ref 30 Parasuraman, P.; Antony, A.P.; B, S.L.S.; Sharan, A.; Siddhardha, B.; Kasinathan, K.; Bahkali, N.A.; Dawoud, T.M.S.; Syed, A. Antimicrobial Photodynamic Activity of Toluidine Blue Encapsulated in Mesoporous Silica Nanoparticles against Pseudomonas Aeruginosa and Staphylococcus Aureus. Biofouling 2019, 35, 89–103, doi:10.1080/08927014.2019.1570501).

Point 2 Continuation of Reviewer comment:

I do not see clearly the advantage of using clay compared to other inorganic matrices (e.g., mesoporous silica, zeolites, etc.) to host a photosensitizer. They mention some examples but I do not clearly see the advantage of using saponite. They claim that one advantage would be its antimicrobial nature (include ref 35). But it is not clearly explained.

Response to the reviewer:

Systems based on layered particles have some advantages. The anchoring of the photosensitizer molecules on the silicate particles allow a suitable control of the distribution of the molecules as well as some photo-physical properties. Prospectively a combination of silicate particles with photoactive molecules and the use of phenomena such as resonant energy transfer, could improve the efficiency of PDT. One of the reason, layered silicates are advantageous, is their compatibility (after an appropriate modification) with some polymeric substances for the synthesis of polymer nanocomposites. The other is the protection of dye molecules. To understand the advantages of our type of material, we would point to some recent review articles (see bellow) that touch on some of the properties of such materials, from which some advantages are also obvious. As our motivation and future plans are not other types of hybrid materials that have already been published on a similar purpose, we only shortly mentioned some of them in Introduction.

Review articles:

Bujdák, J. The Effects of Layered Nanoparticles and Their Properties on the Molecular Aggregation of Organic Dyes. J. Photochem. Photobiol.  C 2018, 35, 108-133.

Bujdák, J., Hybrids with Photoactive Dyes. In Inorganic Nanosheets and Nanosheet-Based Materials: Fundamentals and Applications of Two-Dimensional Systems, Nakato, T.; Kawamata, J.; Takagi, S., Eds. Springer Japan: 2017.

Bujdák J. (2020) Resonance Energy Transfer in Hybrid Systems of Photoactive Dye Molecules and Layered Inorganics. In: Martínez-Martínez V., López Arbeloa F. (eds) Dyes and Photoactive Molecules in Microporous Systems. Structure and Bonding, vol 183. Springer, Cham. https://doi.org/10.1007/430_2020_55

Point 3 Reviewer comment:

The reduction of the photobleaching due to the intercalation of the photosensitizer in the clay is something to look into it. Does the clay reduce the photobleaching?

Response to the reviewer:

We have proven the stabilization of another dye, methylene blue (Lackovičová et al. The Chemical Stabilization of Methylene Blue in Colloidal Dispersions of Smectites. Appl. Clay Sci. 2019, 181.), and some other reports on a similar issue have been published. We have not investigated the stabilization of PhB. On the other hand, the dye properties did not deteriorate after adsorption on the modified Sap film. It is necessary to take into account that we worked with a solid material, in which bacteria were in contact with limited surface area. This is a reason, why a lower concentration of PS is usually more efficient (also after PDT), when PS is tested alone in liquid system or in colloidal dispersion compared to the film on solid material. In design of film fixed on solid material, a question of how much PS could be immobilized to obtain a stable system with optimal physico-chemical properties and maximal antimicrobial properties (while maintaining biosafety) is critical and balance is necessary. And, according to our experiences with testing our system, clay did not reduce the photobleaching. 

Point 4 Reviewer comment:

If the interaction between the modified clay and the photosensitizer is merely electrostatic, only one monolayer of photosensitizer would be adsorbed (bimolecular layers on both surfaces). Does the clay present micropores? Can the photosensitizer be adsorbed on the micropores?

Response to the reviewer:

The space between the layers is primarily filled by alkylammonium cations which were used for the modification of Sap. There is much less of PhB molecules, which are dispersed in the organic phase. We do not assume purely electrostatic interactions. Surface activity of surfactant-modified silicates have potential to adsorb organic molecules in general, including non-ionic ones. Micropores may form secondarily as a result of imperfect arrangement of the individual layers as they form films or larger agglomerates. The presence of micropores is not a typical structural feature in such systems.

Point 5 Reviewer comment:

The main limitation I observe is the limited reduction in the cellular viability after long irradiation times (2.5h) also the toxicity of the PhB itself does not justify the need of using light. It is not clear why you need to apply light when the photosensitizer by itself already shows large antimicrobial action. The benefit shown is reduced.

Response to the reviewer:

Photosensitizer often exhibits a certain bioactivity also in the dark. The aim of this research was to use a generally available cheap light source, such as green LED lights, while 2.5 h of irradiation is not a long period of irradiation (time was estimated on our previous experiences as well as published reports and with agreement of knowledge on “optimal” fluence – J/cm2). The aim of this method is a preparation of biomedical devices using hybrid system as PhB alone cannot be used in equipment. Here, Sap was used as it is generally regarded as a safe, cheap, easy availability, incorporating PhB, and treating with PDI. In Fig 7, it is shown that entrapped PhB in dark did not significantly show a reduction in cell growth and when PDI assisted, growth was significant reduced; thus significance of PDI was proved. It is true, that in the clinical isolate L18, a contribution of PDI was lower, compared to other clinical isolate and standard strain. However, it is not possible to expect similar effects, as each isolate is different (manifesting a different level of response to produced ROS during PDI). As we have experiences that clinical isolates can manifest different level of susceptibility to PDI, we involved in our study not only one standard strain, but two different clinical S. aureus isolates. However, in any way, a contribution of PDI to antimicrobial activity was proved.

Point 6 Reviewer comment:

Do the values reported for the contact angles of sapionite match others reported in the literature?

Response to the reviewer:

Yes, the contact angles of saponite matches with the results of the literature, as the WCA of saponite films have shown a highly hydrophilic nature. See the reference below:

Wu, C.-N.; Saito, T.; Yang, Q.; Fukuzumi, H.; Isogai, A. Increase in the Water Contact Angle of Composite Film Surfaces Caused by the Assembly of Hydrophilic Nanocellulose Fibrils and Nanoclay Platelets. ACS Appl. Mater. Interfaces 2014, 6, 12707-12712.

Point 7 Reviewer comment:

In the photodynamic antibacterial studies as control, they should have used sapionite modified with ODTMA bromide solutions also to decouple the cytotoxic effect of the ODTMA bromide from the cytotoxic effect of the PhB.

Response to the reviewer:

ODTMA was used as a surface modifier to prepare a material to be able to intercalate PhB. No free ODTMA was present; the unbound ODTMA was washed away using Isopropanol and water. To determine antimicrobial activity correctly, we also tested system in dark condition. Differences in antimicrobial activity between dark and light condition significantly proved an enhancement of antimicrobial efficiency in the experiments with irradiation; thus, we do not anticipate a significant effect of ODTA, if so, only negligible.

Point 8 Reviewer comment:

A clear dose-dependent effect is not observed for the illuminated samples in Figure 6. They should also test a concentration above 0.1mM.

Response to the reviewer:

In PDI, a choice of an appropriate concentration of photosensitizer is critical. There is important to find a balance between a concentration that should manifest antimicrobial property (respecting biocompatibility that is another parameter which is tested) and demonstrable PDI effect. If concentration of photosensitizer is very high, a light is not able to penetrate into sample resulting in decreased PDI effect (decrease difference between dark and light samples). Fig. 6 showed that concentration of 0.1 mM is also efficient in dark. Therefore, we selected a concentration of 0.05 mM for a material preparation that showed to be the last suitable for PDI. In this work, we wanted to prepare an efficient material by using minimal amount of PhB with a good effectivity. Obtained log reductions were very promising. Currently, we are preparing polyurethane material modified with film described in this work. Preliminary results confirmed that a higher concentration of PhB is not optimal because of film stability.